# The Severity of Obesity Promotes Greater Dehydration in Children: Preliminary Results

**DOI:** 10.3390/nu14235150

**Published:** 2022-12-03

**Authors:** Agnieszka Kozioł-Kozakowska, Małgorzata Wójcik, Anna Stochel-Gaudyn, Ewa Szczudlik, Agnieszka Suder, Beata Piórecka

**Affiliations:** 1Department of Pediatrics, Gastroenterology and Nutrition, Pediatric Institute, Faculty of Medicine, Jagiellonian University Medical College, 30-663 Krakow, Poland; 2Department of Pediatric and Adolescents Endocrinology, Pediatric Institute, Faculty of Medicine,, Jagiellonian University Medical College, 30-663 Krakow, Poland; 3Department of Anatomy, Institute of Basic Sciences, Faculty of Motor Rehabilitation, University of Physical Education in Cracow, 31-571 Krakow, Poland; 4Department of Nutrition and Drug Research, Institute of Public Health, Faculty of Health Science, Jagiellonian University Medical College, 31-531 Krakow, Poland

**Keywords:** obesity, children’s nutrition, nutritional status, dehydration, diet, fluid intake

## Abstract

The state of hydration of the body depends on the balance between the amount of water and salt consumed and excreted (the urinary extraction of excess sodium requires water). Inappropriate nutrition, particularly consuming too much processed food, causes obesity in children and additionally causes excessive sodium consumption, thus increasing the risk of excessive water loss. The aim of this study was to assess the hydration status of children with obesity and the relation between hydration, body composition, urinary sodium extraction, and nutrient intake. The study group consisted of 27 patients with obesity, with a mean age of 12.89 ± SD 2.79. Each patient’s height, weight, body composition (electrical bioimpedance (BIA)), diet (7-day record), and biochemical tests were assessed. The hydration status was assessed using 24-hour urine collection, 24-hour urine osmolality, and an ultrasound of the vena cava (IVC/Ao index). Overall, 55% of children (*n* = 15) had urine osmolality values above 800 mOsm/kgH_2_O, which indicates significant dehydration, and 53% (*n* = 14) were dehydrated, based on the IVC/Ao index. Children with obesity and dehydration had a significantly higher BMI (31.79 vs. 27.32; *p* = 0.0228), fat mass percentage (37.23% vs. 30.07% *p* = 0.0051), and fat mass in kg (30.89 vs. 20.55; *p* = 0.0158), and significantly higher sodium intake from their diet (3390.0 mg vs. 2921.0 mg; *p* = 0.0230), as well as their sodium/potassium ratio (2.4 vs. 2.0; *p* = 0.0043). The 24-hour urinary sodium excretion and osmolality values were directly related to fat-mass percentage and fat-mass (in kg) in a simple linear correlation analysis. Our preliminary results confirm that obesity is related to dehydration. The overall high sodium excretion in children with obesity indicates an excessive salt intake along with low potassium intake, which is a significant predictor of dehydration, regardless of the total water intake (TWI).

## 1. Introduction

Obesity is a multifactorial chronic disease that may develop at any age, even in early childhood. The number of obese children increased drastically in the years from 1975 to 2016; in 2016, it amounted to 42 million [1]. In 2021, a study conducted by the Institute of the Mother and Child in Warsaw, analyzing a group of 2000 8-year-olds, revealed obesity in 16.8% of the cohort, (19.7% in boys and 13.9% in girls) showing that during the pandemic, the percentage of children who were overweight among these 8-year-olds increased by 5% [2]. Many studies have documented the adverse effects of obesity on the development of children and adolescents. Obesity causes numerous medical, psychological, and social complications [3,4,5,6,7]. Less is known about the relationship between obesity and hydration status in children. The hydration status of the body is mainly influenced by drinks and food intake. Water requirements in children are related to their body mass; their requirements are higher than those of adults, which is partly explained by the physiological nature of the water balance in children, i.e., maturation of the kidneys at about the age of 2, or a higher body surface to body mass ratio, which results in higher water loss through the skin. Additionally, water turnover rates increase with the BMI, as this is based on higher energy requirements, greater food consumption, and higher metabolic production [8]. Mild chronic dehydration leads to urolithiasis, constipation, or urinary tract infections [9]. Insufficient hydration can affect not only physical but also mental health. Even slight dehydration negatively affects the mental condition. It leads to an increased feeling of fatigue and reduces mental efficiency by decreasing memory, attention, and response time. It also reduces visual and motor coordination, being especially important during the learning process [10]. The water needs of individuals are influenced by multiple personal factors, such as anthropometric characteristics, sex, and age. An additional issue is the type of fluids consumed that meet the criteria of well-hydrating fluids and do not contribute to the development of obesity at the same time. A systematic review and a meta-analysis of prospective cohort studies and RCTs provided evidence that sugar-sweetened beverages (SSBs) consumption promotes weight gain in both children and adults [11]. The most recent official guidelines for total water intake were published by the European Food Safety Authority (EFSA) in 2010 [12]. References for total water intake include water from both food and water from beverages of all kinds, including tap water and mineral water. The available data suggest that children do not meet the recommended daily fluid intake; more than 80% of children in many European countries drink less water than is recommended in the EFSA guidelines, and 54.5% of children are insufficiently hydrated (see Table 1) [8,13,14,15,16,17,18,19,20,21,22,23]. 

Total water intake (TWI) is very important but is probably not the most important factor influencing dehydration. In fluid balance, a hugely important role is played by the appropriate intake of sodium and potassium from the diet. The excretion of excessive amounts of sodium requires larger amounts of water, while potassium intake also has an effect on sodium excretion [24]. Studies show that too much intake of sodium/salt in child populations and a high amount of salt consumption are significantly associated with obesity in children [25,26].

There are few studies assessing the hydration aspect in overweight or obese children [20,27]. Our previous population studies showed that the hydration level was lower in children with an excessive body fat percentage (BF%) than in children with normal BF% [19]. Given the increasing incidence of childhood obesity and the fact that obesity itself is a risk factor for dehydration, proper hydration becomes a challenge in children with obesity. The aim of this study was to assess the hydration status of children with obesity and the association between hydration, body composition, urinary sodium extraction, and nutrient intake. 

## 2. Materials and Methods

Recruitment for the study lasted from October 2021 to April 2022. The study group consisted of 34 pubertal children with obesity (a BMI over 90 centile compared to national growth charts) who were patients of the University Children’s Hospital in Krakow, Poland. The following exclusion criteria were established: chronic kidney disease, heart disease, diabetes mellitus, and other overt endocrinopathies with a possible impact on water balance, and a lack of consent. Finally, 27 children (20 girls, 17 boys), were enrolled for further analyses; the age range was 10.1–17.9 (mean age 12.89 ± SD 2.79). Study group and data collection was shown on Figure 1. The protocol of the study was conducted according to the ethical principles stated in the Helsinki Declaration (1964). All procedures involving the research-study participants were approved by the Bioethical Committee of the Jagiellonian University, No. 10.72.6120.235.2020. Written informed consent was obtained from all patients.

### 2.1. Data Collection

The children and their parents were asked to give their consent to participate in the study; they were informed about the need to make a current dietary record and instructed on the method used for daily urine collection. At the next visit to the hospital, the parents returned food diary records from 7 days and urine samples from the 24-hour urine collection process, along with blood tests. An ultrasound of the vena cava and body composition assessment (BIA) were then performed. After that, a dietary consultation was conducted. 

### 2.2. Anthropometric Parameters

In the study group, body weight and height were measured to the nearest 0.1 kg and 0.1 cm, respectively, using a stadiometer (SECA). As the standard of reference, the normal values of the local population were used. In order to assess nutritional status, body mass index (BMI) was calculated according to the formula: BMI = weight (kg)/height^2^ (m^2^). The BMI Z-score (BMI-z) was identified and interpreted in relation to national percentile charts [28]. 

The body composition of the studied children was assessed via bioelectrical impedance analysis (BIA), using a multi-frequency bioelectrical impedance analyzer (Tanita BC 418 S MA, Tokyo, Japan). The measurements were performed according to the manufacturer’s guidelines at least 2 h after the ingestion of a light breakfast and urination. The following data were collected—fat mass (FM kg), fat percentage of the whole body (fat mass %), fat-free mass (FFM) (in kg), and total body water (TBW) (in kg).

### 2.3. Biochemical Assessment and Blood Pressure Examinations

Creatinine, total cholesterol, triglycerides, high density cholesterol HDL, low density cholesterol LDL, uric acid, glucose (serum 0), glucose (serum 120), and vitamin D3 were determined in the laboratories of the hospital where the tests were conducted, according to standard laboratory procedures, in accordance with good laboratory practice (GLP) and In Vitro Diagnostic/Food and Drug Administration (IVD/FDA) standards. Blood pressure measurement was performed by auscultation, using a standard blood pressure monitor with a cuff. The measurements were repeated three times. Systolic blood pressure (SBP) and diastolic blood pressure (DBP) were calculated as the mean value of these three outcomes [29].

### 2.4. Dehydration Assessment

The patients were trained by the nurse on how to collect urine for the whole day. Collection of the urine always took place on the day preceding the visit to the hospital, during which visit the subsequent examinations were carried out. After collection, all urine samples were stored at +4 °C before the contents were determined. The urine volume, 24-hour urine osmolality (mOsm/kgH_2_O), 24-hour urinary sodium excretion (mmol/24 h), 24-hour urinary sodium extraction (mg/24 h), and 24-hour urine sodium concentration (mmol/L) were determined. An ultrasound assessment of the inferior vena cava, as well as the abdominal aorta, was performed in all patients using a Hitachi Aloka Arietta 70 with 12 1–5 MHz probes. The width of the inferior vena cava (IVC) depends on hydration status, whereas the abdominal aorta (Ao) width does not. In addition, the abdominal aorta width correlates with age, sex, and body surface area. Thus, the IVC/Ao index was used to assess the hydration status of the study population. An IVC/Ao index value of 1.2 +/− 2 SD, SD = 0.17 indicates optimal hydration status. A Polish Ultrasound Society-certified operator performed the tests [30]. The doctor performing the examination was blinded to the clinical and laboratory data of the patients.

### 2.5. Dietary Assessment

Everything that the child ate and drank on a given day was recorded by the parents or children (depending on age) on an ongoing basis for 7 days before the date of the visit. The parents also filled in a nutrition questionnaire that contained questions about food and beverage consumption, frequency, and type. The Aliant dietetic calculator was used to evaluate the implementation of nutrition standards. Dietary supplements were included in the nutrient calculations. To evaluate the prevalence of nutrient adequacy, the estimated energy requirement (EER) for energy, the recommended daily allowances (RDA) for protein, the recommended intake (RI) for fats and carbohydrates, and adequate intake (AI) were used. The values of the macro- and micronutrients consumed were checked with the Polish norm, according to the age and sex of the child.

### 2.6. Statistics

Dietetic software was used for the nutrient analysis of the food records, according to the Polish National Food and Nutrition Institute database [31]. The study group was divided up, based on urine osmolality, as follows: optimal hydration urine osmolality of ≤500 mOsm/kgH_2_O and ≤800 mOsm/kgH_2_O, with dehydration urine osmolality of >800 mOsm/kgH_2_O. All the collected data were analyzed statistically with the use of Statistica 13.0 software (StatSoft). The following descriptive statistics were calculated: mean, standard deviation, median, and percentages. The Shapiro–Wilk statistic test for testing the normality was applied. Student’s *t*-test was used to check the association between hydration level and the medical and anthropometric parameters. Spearman correlation coefficients were employed to check the relationship between 24-hour urinary sodium extraction, 24-hour urine osmolality, and the medical and anthropometric parameters. An analysis of variance was used to assess the relationship between the dependent variable (osmolality) and the independent variables. The significance level was set at *p* < 0.05.

## 3. Results

The mean value of the BMI Z-score was 3.36 (SD 1.2). In the study group, none of the patients had previously had a consultation with a dietitian and did not have a modified diet. Overall, 55% of children (*n* = 15) had urine osmolality values above mOsm/kgH_2_O, which indicates significant dehydration, and 53% (*n* = 14) were dehydrated, based on the IVC/Ao index. Children with obesity consumed, on average, 33% more energy than was recommended. In terms of all the macronutrients consumed, the consumption standards were exceeded, mostly from simple sugars. Total water intake from the diet was not consistent with the recommendations (see Table 2). 

The average percentage of water from liquids and food in the study group is as follows: plain water (44%), tea (17%), soup (12%), SSBs (10%), 100% fruit juice (8%), milk and yogurt (5%), and flavored water (4%). 

The most common concomitant problem in the study group was vitamin D deficiency (*n* = 20, 76%). Impaired fasting glucose was observed in one patient (5%), while impaired glucose tolerance was diagnosed in 5 (19%), elevated triglycerides in 6 (22%), low HDL cholesterol in 4 (14%), and high uric acid in 12 (45%). All patients had an appropriate creatinine level. There were no differences between the biochemical variables, depending on the dehydration status. In the group of children with dehydration, a significantly higher systolic blood pressure level was observed (*p* = 0.0355) (see Table 3).

Table 4 shows data from the 24-hour urine samples and the anthropometric tests. The level of urine volume was between 500 and 2500 mL, with a mean value of 1444.165 mL/24 h, and did not change depending on hydration status. The mean measured urinary sodium excretion was 3730.99 mg/day, and the mean sodium intake was 2823.67 (915.43) mg/day. Children with obesity and dehydration had significantly higher sodium intake from their diet (3390.0 mg vs. 2921.0 mg, *p* = 0.0230) equal to their salt intake (8.47 vs. 7.27 g/day), as well as the sodium/potassium ratio (2.40 vs. 2.03; *p* = 0.0043). There were no differences in other urinary results, depending on the level of hydration in the studied group of obese children.

Children with obesity and dehydration had significantly higher BMI (31.79 vs. 27.32; *p* = 0.0228), fat mass % (37.23% vs. 30.07%; *p* = 0.0051) and fat mass in kg (30.89 vs. 20.55; *p* = 0.0158). The 24-hour urinary sodium excretion and 24-hour urine osmolality were directly related to fat mass (%) and fat mass (kg) in a simple linear correlation analysis (Table 4). The 24-hour urine osmolality value was also correlated with TBW %. Neither the 24-hour urine osmolality nor the 24-hour urinary sodium extraction was related to TWI. In the same analysis, systolic BP was also directly related to 24-hour urinary sodium excretion (Table 5). 

Regression analysis was used to estimate the influence of individual independent variables on the level of 24 h urine osmolality. The model was statistically significant (F(16.47) = 25.760; *p* < 0.001) and all predictors accounted for a total of 75% of the dependent variable (R = 0.75). A significant influence on the level of 24-hour urine osmolality was represented by the daily intake of sodium (β = 0.83; *p* < 0.001), daily potassium intake (−0.48; *p* < 0.001), and fat mass % (β = 0.34; *p* < 0.001). TWI turned out to be a non-existent predictor.

## 4. Discussion

### 4.1. Fluids Intake 

In the UK National Diet and Nutrition Survey, which examined the consumption of beverages in a group of 854 children aged 4–13, the percentage of children who failed to meet EFSA guidelines was 88.7% (84.4% for 4–8-year-old group and 92.8% for the 9–13-year-old group). Overweight children consumed the most water from various drinks, but this mainly came from sweetened drinks and sugary carbonated drinks [32]. In our study, 44% of TWI came from plain water and the rest came from other beverages and foods. In the review by Suh and Kavouras, the TWI of children from 25 countries was assessed. Among the 19 countries that reported children’s water intake, 60 ± 24% of children failed to meet the hydration recommendations [33].

### 4.2. Hydration Status and Obesity

Dehydration can be observed in children in the general population; according to a study conducted in Belgium in a group of 371 children aged 7–13, based on urine osmolality (>800 mOsm/kgH_2_O), dehydration was observed in 76% of children (morning urine sample) and 54% of children (school day urine sample) [21]. Hydration status has proved to be problematic, particularly in boys and obese children. In a survey carried out in a group of 264 children aged 7–15 years from Poland, insufficient hydration during a school day was identified in 53% of children, with severe dehydration in 16.3% (>1000 mOsm/kgH_2_O). This higher level of dehydration was observed in children with obesity. Children with a high body fat percentage (BF%) showed a 2.3-fold increase in the odds of dehydration during a school day [19]. In a study by Maffeis et al., hydration levels were assessed in 86 obese and 89 healthy children aged 7–11 years. Children with obesity were less hydrated than healthy children because, given their relative BMI-z score, they drank less [20]. In a cross-sectional study, children aged between 7 and 18 years with obesity (*n* = 31) were compared with non-obese children (*n* = 30). Children with obesity had lower fluid consumption, lower total body water percentages, and higher urine density, and were less hydrated than normal-weight children. The drinking of SSBs was at 71% in children with obesity and at 20% in healthy volunteers [27]. 

### 4.3. Sodium Intake/Sodium Extraction and Obesity

According to the position taken by the Panel on Nutrition of the European Food Safety Authority (EFSA), a sodium intake considered safe and appropriate for children (11–17 years) corresponds to the value for adults, adjusted according to the energy demand and an increase of 2000 mg/day for children aged 11–17 years [34]. There are no established recommendations for pediatric patients with obesity. In the general population, sodium intake is significantly different from the recommended intake. The median sodium intake in the pediatric population (4–17 years) between 2003 and 2016 was 2840 mg/day (95% CI, 2805–2875 mg/day), decreasing from 2912 mg/day (95% CI 2848–2961 mg/day) in 2003–2004 to 2787 mg/day (95% CI, 2677–2867 mg/day) in 2015–2016 (NHANES) [26]. In Europe, the situation is very similar. In Portugal, 83% of adolescents had a sodium intake higher than what was recommended, while in France, the pediatric population’s median sodium intake was 2245 mg/day [35,36]. In Asia, a study conducted among Iranian children showed a daily sodium intake of 3160.0 mg/day, corresponding to a salt intake of 7.9 g/day [37]. These results were near to the estimated sodium intake presented in our study. The mean value of sodium excretion in adolescents, based on 24-hour sodium excretion, was 3072 mg/day in Portugal, 2967 mg/day in Italy, 3270.6 mg/day in Spain, 3401.47 mg/day in England, 3013 mg/day in Germany, and 3130 mg/day in Iran [35,38,39,40,41]. In our study, the mean value of sodium excretion was slightly higher, at 3730.99 mg/day. A higher value of sodium was excreted than was consumed. However, it is difficult to assess the intake of sodium or salt in the diet, using the available tools, even if the amount in the products consumed was accurately recorded; patients may not record the exact amount of salt that they add to their dishes. In addition, the amount of salt in the finished products is not always visible on the labels.

Increased salt consumption is one of the contributing factors for childhood obesity, which results from the increased consumption of processed food [42]. Salt preference and eating habits are established during childhood and are often maintained until adulthood [43]. Parents have a great influence on the formation of children’s taste preferences and eating habits, and this is one of the main reasons why parental obesity is positively associated with the obesity of their offspring [44].

The Korean National Health and Nutrition Examination Survey have investigated correlations between childhood obesity and the parents’ or children’s sodium intake. The BMI-z score and 24-hour urinary sodium excretion were used to examine the associations between obesity and sodium intake. A close relationship was observed between obesity in children and their sodium intake, which also correlated well with parental BMIs and diet behavior. Children with higher urinary sodium excretion showed higher BMI and higher parental obesity, compared to those with lower urinary sodium excretion; however, the statistical significance of the latter relationship varied according to sex [45]. 

Sodium intake was positively associated with being overweight and obese and central to obesity among US children, regardless of energy and SSBs intake, but this association was not statistically significant in adolescents. Using a US nationally representative survey, NHANES 2009–2016, it was found that a higher daily sodium intake was associated with increased odds of being overweight and/or obese, irrespective of the intake of energy and SSBs among US children. Researchers observed a positive association between the estimated sodium intake and central obesity (WC > 90th percentile in children) [46].

High salt intake was also significantly associated with an excessive level of body fat in both children and adults in the UK National Diet and Nutrition Survey. It was observed that salt intake, assessed on the basis of 24-hour urinary sodium excretion, was higher in overweight and obese people. The study found that an increase in sodium intake by 1 g daily was associated with a 28% higher risk of obesity in the pediatric population [26].

In the study by Grimes et al., in a group of 6400 children aged 10.1 ± 0.1 years, the mean sodium intake was 3056 ± 48 mg/d (7.8 ± 0.1 g salt/d) and was positively associated with fluid intake (r = 0.42, *p* < 0.001). After adjustment for age, gender, racial group, SES, and BMI, each additional 390 mg of Na/d (1 g salt/d) was associated with a 74 g/d higher fluid intake (*p* < 0.001). In those subjects consuming SSB (*n* = 4443; 64%), each additional 390 mg of Na/d (1 g salt/d) was associated with a 32 g/d higher SSBs intake (*p* < 0.001) [47].

In the MINISAL study in the pediatric population, a gradual increase in the estimated sodium intake was observed with an increase in the BMI Z-score, which supports the additive contribution of obesity to higher sodium intake. Sodium intake, based on 24-hour urinary excretion, was positively associated with adiposity in children from Iran. In the study group (11–18 years), those with the highest sodium excretion had a higher chance of obesity (excessive fat tissue) compared to those in the lowest category (OR: 1.79; 95% CI: 1.08–2.74) [38]. In our study, it was confirmed that children with obesity consume too much sodium, while children with a higher level of body fat percentage had higher urinary sodium extraction, which proves the higher sodium intake along with the degree of obesity. What is more, a high sodium intake promotes the faster accumulation of body fat [25]. In experimental studies, it was noted that a higher intake of Na is associated with increased lipogenic activity and fat formation. Moreover, glucose uptake and its conversion to lipids within the adipocytes was higher in mice consuming a higher amount of sodium [48]. These results are worrying, especially considering that obesity in adolescents is associated with greater sensitivity to salt in terms of blood pressure. In our study, systolic BP was directly related to 24-hour urinary sodium excretion. Many of the processes leading to the development of hypertension that are associated with excess body fat and salt intake resemble a vicious circle [24]. High potassium intake and a low Na/K ratio also seem to positively affect blood pressure in childhood. An increased potassium intake from the diet can balance out excessive sodium intake. The WHO recommends a potassium intake of at least 3510 mg/day. A diet rich in fruit and vegetables helps to fulfill these recommendations. Adequate potassium intake in the diet is very important for lowering blood pressure, but excessive potassium supplementation should be avoided [49,50]. The main limitation of the paper is a small sample size, therefore yielding limited statistical power for the study. The issue requires further research using larger groups. 

## 5. Conclusions

Our preliminary results confirm that the severity of obesity promotes greater dehydration in children. This study also confirms that high sodium excretion in obese children indicates an excessive salt intake, along with a low potassium intake, which is a significant predictor of dehydration, regardless of the TWI. Dietary intervention in children with obesity should focus on the higher consumption of potassium-rich foods, mainly vegetables and fruits, in order to compensate for the effect of sodium increase, as well as avoiding sodium-rich foods. Regardless of this intervention, children should be educated about the correct amount of water consumption.

## Figures and Tables

**Figure 1 nutrients-14-05150-f001:**
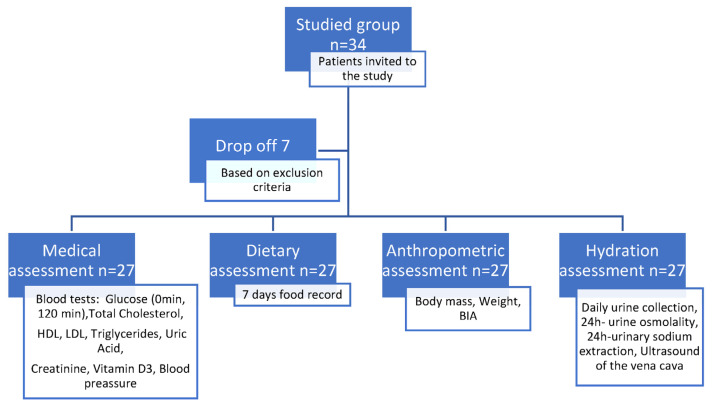
Study group and data collection. h: hours.

**Table 1 nutrients-14-05150-t001:** Cross-sectional, longitudinal, and intervention studies including hydration status measurement in children.

References	Studied Group	Country	Type of the Study and Year	Methods	Outcome	Results
Alexy et al. (2012) [14]	Children and adolescents (4–18 years old)	Germany	Cross-sectional study, 2003–2009	Three-day nutrition diary of weighed foods and nutrients	Hydration state based on 24-hour sodium excretion	Salt excretion increased with age but remained constant during the study
Bonnet et al. (2012) [15]	Children (9–11 years old)	France	Cross-sectional study, 2010	Food diary of breakfast	Water intake and hydration status at breakfast	More than a third of the children had high urine osmolality (801 and 1000 mmol kg^−1^) and total water intake was inversely correlated with high urine osmolality
Bougatsas et al. (2018) [16]	Children (8–14 years old)	Greece	Cross-sectional study 2013	Fluid intake from two days	Assessment of fluid consumption and hydration level	Drinking water and milk was correlated with better hydration status, whereas drinking regular soda and other drinks was associated with worse hydration (*p* = 0.001)
Kavouras et al. (2016) [17]	Children (8–14 years old)	Greece	Cross-sectional study, 2012–2013	Color of urine	Assessment of urine color	The traditional urine color score scale is a good method for assessing children’s hydration; the average 24-hour urine color was 3 and the average 24-hour urine osmolality was 686 mmol kg^−1^
Kavouras et al. (2017) [18]	Children (9–13 years old)	Greece	Cross-sectional study, 2012–2013	Food diary of fluid intake from 2 day	Assessment of daily water intake and biomarkers of hydration	Insufficient water intake from fluids was associated with lower levels of hydration in children
Kozioł-Kozakowska at al. [19]	Children (7–15 years old)	Poland	Cross-sectional, 2018	Urine osmolality during a school day	Hydration state	In all, 53% of the children were insufficiently hydrated, and 16.3% of them had urine osmolality > 1000 mOsm/kgH_2_O, which indicates severe dehydration
Maffeis et al. (2016) [20]	Children (7–11 years old)	Italy	Cross-sectional study	Three-day nutrition diary of weighed foods and fluid	Assessment of fluid intake and level of hydration in obese children vs. normal weight	Obese children were less hydrated compared to normal-weight children and drink less when considering body mass index
Michels et al. (2017) [21]	School children (7–13 years old)	Belgium	Cross-sectional study, 2014	Food frequency questionnaire FFQ for children	Assessment of children’s hydration at school and its predictors	The children had a high risk of dehydration (school urine osmolality of 888 mmol kg^−1^); however, this was not due to the quality of the diet
.Padrão et al. (2016) [22]	Children (7–11 years old)	Portugal	Cross-sectional study, 2014	Twenty-four-hour dietary interview	Assessment of diet and biomarkers of hydration	More than half of the children had too low hydration levels; higher water intake was associated with better hydration levels
Stahl et al. (2007) [23]	Children (4–11 years old)	Germany	Cross-sectional study, 2007	Three-day nutrition diary of weighed foods	Assessment of the relationship between hydration and diet	Children who were properly hydrated had a higher total water intake with the diet and lower energy density of the diet, compared with children with lower hydration levels
Stookey et al. (2012) [8]	Children (9–11 years old)	USA	Cross-sectional study, 2009	Food diary for breakfasts	Assessment of hydration and intake of water	Elevated urinary os-molarity (>800 mmol kg^−1^) was associated with lower water intake in the morning

**Table 2 nutrients-14-05150-t002:** Daily energy, macronutrients, TWI intake, and norm implementation in the studied group.

	Daily Intake g/day X (SD)	% of Energy X (SD)	% Norm Implementation X (SD)	Recommended Intake
Energy				
kcal/day	2871.33 (676.35)	62.42 (13.60)	133 (21.11) % EER	1800–2450 kcal/day depending on weight, age, and sex (7–15 years old)
Protein				
g/day	86.6 (33.9)	12.20 (4.7)	102.1 (16.04) % RDA	10–20% Energy
Carbohydrates				
g/day	423.30 (127.8)	59.1 (4.6)	117.0 (45.29) % RI	55–60% Energy
Sugars				
g/day	100.46 (29.48)	15.2 (3.1)	275.1 (95.68) % RI	Less than 10% simple sugar for energy
Fats				
g/day	111.6 (39.4)	35.1 (14.1)	112.6 (31.82) % RI	25–30% Energy
TWI				
ml/day	2127.31 (633.93)	-	97% RI	2–2.5 L

Estimated energy requirement (EER) for energy, recommended daily allowances (RDA) for protein, reference intake (RI) for fats and carbohydrates, adequate intake (AI) for water, water from food, and all liquids (TWI).

**Table 3 nutrients-14-05150-t003:** Metabolic parameters and blood pressure in the study group, depending on the hydration level.

Parameters	Total *n* = 27	Proper Hydration *n* = 12	Dehydration *n* = 15	Student’s *t*-Test *p*-Value
Fasting glucose (mmol/L)	4.81 (0.26)	4.8 (0.30)	4.8 (0.22)	0.4050
Glucose 120′ (oral glucose tolerance test) (mmol/L)	5.48 (0.66)	5.4 (0.61)	5.5 (0.71)	0.2672
Triglycerides (mmol/L)	1.30 (0.66)	4.0 (0.91)	4.5 (0.91)	0.1571
Total cholesterol (mmol/L)	4.26 (0.91)	4.0 (0.92)	4.5 (0.90)	0.8460
HDL cholesterol (mmol/L)	1.25 (0.22)	1.27 (0.15)	1.24 (0.27)	0.1741
LDL cholesterol (mmol/L)	2.51 (0.81)	2.27 (0.85)	2.69 (0.79)	0.5982
Creatinine	49.41 (11.95)	45.6 (5.12)	51.7 (14.90)	0.8896
25(OH)D (ng/mL)	21.67 (8.32)	21.7 (9.50)	22.2 (7.73)	0.1266
Uric acid (µmol/L)	334.24 (83.48)	325.22 (95.12)	336.99 (79.52)	0.0792
Systolic blood pressure (mmHg)	102.2 (8.32)	101.8 (8.70)	105.1 (10.21)	0.0355
Diastolic blood pressure (mmHg)	63.2 (4.61)	61.1 (3.30)	63.3 (5.10)	0.9295

Values are presented as mean and standard deviations (SD); a *p*-value of less than 0.05 was considered significant.

**Table 4 nutrients-14-05150-t004:** Differences in body composition, urine tests, and sodium and potassium intake, depending on hydration status.

Parameters	Total *n* = 27	Proper Hydration *n* = 12	Dehydration *n* = 15	Student’s *t*-Test *p*-Value
Age (years)	12.89 (2.79)	12.27 (2.58)	13.32 (3.02)	0.4706
Weight (kg)	77.39 (25.44)	69.39 (24.38)	81.87 (26.36)	0.2441
Height (cm)	157.73 (17.81)	156.70 (18.83)	158.07 (18.33)	0.7391
BMI (kg/m^2^)	30.15 (5.08)	27.32 (3.59)	31.79 (5.30)	0.0228
Fat mass (kg)	26.63 (11.53)	20.55 (6.58)	30.89 (12.78)	0.0158
Fat mass (% of weight)	34.05 (6.59)	30.07 (3.75)	37.23 (6.49)	0.0051
Fat-Free Mass (kg)	49.67 (18.10)	48.34 (18.32)	49.42 (18.64)	0.9116
Muscle Mass (kg)	47.80 (16.17)	45.91 (7.45)	48.03 (15.86)	0.8244
TBW (kg)	36.84 (12.41)	35.3 (13.38)	37.03 (12.16)	0.8028
TBW (% of body weight)	48.33 (5.23)	51.09 (4.20)	46.12 (4.95)	0.0265
Urine volume (mL/24 h)	1444.1 (500.43)	1472.9 (585.3)	1339.0 (338.82)	0.6353
24-hour sodium concentration (mmol/L)	133.59 (49.62)	109.47 (41.57)	152.36 (48.48)	0.0508
24-hour urinary sodium excretion (mmol/24 h)	170.91 (85.16)	156.11(44.30)	178.47 (103.65)	0.8973
24-hour urinary sodium excretion (mg/24 h)	3730.99 (1958.57)	3590.6 (1019.00)	3804.7 (2384.01)	0.8973
Sodium intake (mg/24)	2823.67 (915.43)	2921.0 (557.00)	3390.0 (692.02)	0.0230
Potassium intake (mg/24)	1421.65 (1172.41)	1438 (1637.01)	1408.1 (741.00)	0.1655
Sodium/potassium ratio	1.98 (0.47)	2.03 (0.30)	2.40 (0.42)	0.0043

Values are presented as mean and standard deviations (SD), *p*-value of less than 0.05 was considered significant, TBW—total body water.

**Table 5 nutrients-14-05150-t005:** Simple linear correlation analysis: 24 h urinary sodium, 24 h urine osmolality, and specified variables (Spearman correlation coefficients (rS)).

24-hour Urinary Sodium Extraction (mg/24 h)						24-hour Urine Osmolality (mOsm/kgH_2_O)					
BMI	Fat Mass (%)	Fat Mass (kg)	TBW%	TWI %	Systolic BP	BMI	Fat Mass (%)	Fat Mass (kg)	TBW%	TWI	Systolic BP
0.071	0.425 *	0.431 *	0.060	−0.113	0.155^*^	0.343	0.574 **	0.398 *	0.442 *	0.022	0.132

TBW%—total body water, TWI—total water intake * Significant at the 0.05 level (2-tailed), ** Significant at the 0.01 level (2-tailed).

## Data Availability

The data presented in this study are not publicly available for confidentiality reasons. These data are available on request from the corresponding author.

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
