# Peer review of "The Severity of Obesity Promotes Greater Dehydration in Children: Preliminary Results"

_nutrients, 2022, doi:10.3390/nu14235150_

Round 1
Reviewer 1 Report
> 1. What is the main question addressed by the research? The aim of this study was to assess the hydration status of children 18 with obesity and the relation between hydration, body composition, urinary sodium extraction and 19 nutrients intake.
> 2. Do you consider the topic original or relevant in the field? Does it > address a specific gap in the field? Yes.
> 3. What does it add to the subject area compared with other published > material? Need improvements.
> 4. What specific improvements should the authors consider regarding the > methodology? What further controls should be considered? The authors need to address the comments.
> 5. Are the conclusions consistent with the evidence and arguments presented > and do they address the main question posed? Yes.
> 6. Are the references appropriate? Yes.
> 7. Please include any additional comments on the tables and figures. I suggest to the authors add a table in the Introduction section. The table should show the classification of studies assessing the hydration aspect according to the properties and main sources.
The authors should add more detail to assess risk factors and independent variables for dehydration.
There was a one-year time lag between their measurement of children's health outcomes. So, they could not assess seasonality/temporal variability.
The authors did not triangulate dehydration measurements with physical symptoms of dehydration or children's report of thirst.
Reviewer 2 Report
Thank you for your submission on an important topic. I would suggest emphasizing why hydration status is clinically important. Does dehydration lead to secondary health consequences. Also, you suggest increasing potassium intake with fruits and vegetables. Could vitamin supplements achieve the same intended benefit? If not, this should be discussed and presented, please.
Round 2
Reviewer 1 Report
The authors already addressed all the comments. The quality of the article has enhanced based on my comments and other reviewers' comments. Therefore, I don’t have further comments.